

# Accessible molecular phylogenomics at no cost: obtaining 14 new mitogenomes for the ant subfamily Pseudomyrmecinae from public data

Gabriel A. Vieira and Francisco Prosdocimi

Instituto de Bioquímica Médica Leopoldo de Meis, Universidade Federal do Rio de Janeiro, Rio de Janeiro, Rio de Janeiro, Brazil

Corresponding authors
Gabriel A. Vieira,
gabriel.vieira@bioqmed.ufrj.br,
fprosdocimi@gmail.com
Francisco Prosdocimi,
prosdocimi@bioqmed.ufrj.br

## ABSTRACT

The advent of Next Generation Sequencing has reduced sequencing costs and increased genomic projects from a huge amount of organismal taxa, generating an unprecedented amount of genomic datasets publicly available. Often, only a tiny fraction of outstanding relevance of the genomic data produced by researchers is used in their works. This fact allows the data generated to be recycled in further projects worldwide. The assembly of complete mitogenomes is frequently overlooked though it is useful to understand evolutionary relationships among taxa, especially those presenting poor mtDNA sampling at the level of genera and families. This is exactly the case for ants (Hymenoptera:Formicidae) and more specifically for the subfamily Pseudomyrmecinae, a group of arboreal ants with several cases of convergent coevolution without any complete mitochondrial sequence available. In this work, we assembled, annotated and performed comparative genomics analyses of 14 new complete mitochondria from Pseudomyrmecinae species relying solely on public datasets available from the Sequence Read Archive (SRA). We used all complete mitogenomes available for ants to study the gene order conservation and also to generate two phylogenetic trees using both (i) concatenated set of 13 mitochondrial genes and (ii) the whole mitochondrial sequences. Even though the tree topologies diverged subtly from each other (and from previous studies), our results confirm several known relationships and generate new evidences for sister clade classification inside Pseudomyrmecinae clade. We also performed a synteny analysis for Formicidae and identified possible sites in which nucleotidic insertions happened in mitogenomes of pseudomyrmecine ants. Using a data mining/bioinformatics approach, the current work increased the number of complete mitochondrial genomes available for ants from 15 to 29, demonstrating the unique potential of public databases for mitogenomics studies. The wide applications of mitogenomes in research and presence of mitochondrial data in different public dataset types makes the ''no budget mitogenomics'' approach ideal for comprehensive molecular studies, especially for subsampled taxa.

## INTRODUCTION

More than one decade after the advent of next-generation sequencing (NGS) (*Margulies et al., 2005*), it is now clear that this mature technology fostered an unprecedented increase in the generation of genomic data together with an important reduction in sequencing costs (*Mardis, 2008*; *Van Dijk et al., 2014*; *Goodwin, McPherson & McCombie, 2016*). In order to gather and democratize the access to genomic data, the International Nucleotide Sequence Database Collaboration (INSDC, http://www.insdc.org/) has been established in 1987. This continuous effort comprises three international centers: (i) the National Center for Biotechnology Information (NCBI), (ii) the European Bioinformatics Institute (EBI) and (iii) the DNA Data Bank of Japan (DDBJ) (*Karsch-Mizrachi et al., 2017*). As part of this initiative, the Sequence Read Archive (SRA) was created to host raw sequence reads and metadata generated by NGS projects (*Kodama, Shumway & Leinonen, 2012*). Making raw sequence data available is key for the experimental reproducibility (*Stodden, Seiler & Ma, 2018*), a pillar of scientific endeavor. SRA database has been recurrently used to support new research, such as: the evaluation of single nucleotide polymorphisms and deletions (*Bordbari et al., 2017*), the test of new bioinformatics pieces of software (*Simpson et al., 2009*; *Langmead & Salzberg, 2012*; *Bolger, Lohse & Usadel, 2014*), and also to evaluate the impacts of common procedures on data, such as trimming (*Del Fabbro et al., 2013*), among other studies (*Kayal et al., 2015*; *Bernstein, Doan & Dewey, 2017*; *Linard et al., 2018*).

The availability of public data is continuously growing together with the potential uses of such databases to the scientific community. In a 2-year period (August-2015 to August-2017), 3,000 trillion base pairs have been added to SRA, promoting a 233% growth of the repository (*Karsch-Mizrachi et al., 2017*). However, potential uses of these data are far from being fully exploited, once public databases present resources that could be used to address diverse ranges of unexplored biological questions, as pointed out in the previous paragraph. Here we focus in searching for the presence of complete mitochondrial genomes in public genomics datasets.

### Mitogenomes: ubiquity in datasets and relevance in scientific research

Whole-Genome Sequencing (WGS) experiments and partial genome sequencing projects normally yield enough sequencing reads from mitochondria to allow the assembly of complete mitogenomes (*Prosdocimi et al., 2012*; *Smith, 2015*). These small organellar genomes can be often assembled in high coverage due to the high copy number of these organelles (*Smith, 2015*). Also, previous studies indicate that it is possible to recover complete and/or nearly complete mitochondrial sequences from RNA-Seq data (*Tian & Smith, 2016*; *Rauch et al., 2017*; *Plese et al., 2018*) and targeted sequencing strategies as exome (*Picardi & Pesole, 2012*; *Guo et al., 2013*; *Samuels et al., 2013*) and UCE (Ultra Conserved Elements) off-target data (*Raposo do Amaral et al., 2015*; *Miller et al., 2016*). The assembly of numerous complete mitogenomes and/or large mitochondrial contigs from the sequencing of pooled multi-species samples has also been performed

successfully (*Timmermans et al., 2016*; *Linard et al., 2018*) under an approach named 'mito-metagenomics' (*Tang et al., 2014*) or 'mitochondrial metagenomics' (MMG) (*Crampton-Platt et al., 2015*).

Some works have demonstrated the potential of public data to mitogenomic studies by successfully using public data to assemble mitochondrial sequences (*Diroma et al., 2014*; *Kayal et al., 2015*; *Linard et al., 2018*). However, a large number of species that have genomic data available in the SRA database are still lacking works describing their complete mitochondrial sequences.

Due to their small sizes, high conservation and the absence of introns, mitogenomes are the most commonly sequenced chromosomes, especially for metazoans (*Smith, 2015*). Mitochondrial genomes are poorly sampled for many taxa and therefore our current knowledge about evolutionary biology of many clades could be improved with the use of public data. Being primarily maternally inherited and non-recombinant, such sequences are often used to study evolutionary biology (*Finstermeier et al., 2013*; *Krzemińska et al., 2017*), population genetics (Pečnerová et al., 2017; *Kılınç et al., 2018*), phylogeography (*Chang et al., 2017*; *Fields et al., 2018*), systematics (*Lin et al., 2017*; *Crainey et al., 2018*) and conservation (*Moritz, 1994*; *Rubinoff, 2006*; *Rosel et al., 2017*) of various clades (*Avise, 1994*), specially from subsampled taxa (*Gotzek, Clarke & Shoemaker, 2010*; *Duan, Peng & Qian, 2016*) and non-model organisms (*Prosdocimi et al., 2012*; *Tilak et al., 2014*; *Plese et al., 2018*).

## Mitogenome sampling in the Formicidae family

An example of poor mitogenome taxon sampling occurs in ants (Hymenoptera: Formicidae). Despite being an ubiquitous, ecologically dominant and hyper diverse group (*Holldobler & Wilson, 1990*) with over 13,000 species described (*Bolton, 2012*), complete mitogenome records are available for mere 15 species in GenBank.

UCE sequencing data have been previously used to study ant phylogeny (*Blaimer et al., 2015*; *Ward & Branstetter, 2017*; *Branstetter et al., 2017*), but attempts to recover mitochondrial sequences from the off-target data generated are limited (*Ströher et al., 2017*). Thus, these pieces of information have not been used to further understand evolutionary relationships for the clade.

## The Pseudomyrmecinae subfamily: taxonomy, ecology and evolution

One particular ant group that suffers from poor mitogenome sampling is the ant subfamily Pseudomyrmecinae that contains three genera: (i) the New-World genus *Pseudomyrmex*, consisting of ~137 species, most of which can be classified in one of the ten morphological species groups described (*Ward, 1989*; *Ward, 1993*; *Ward, 1999*; *Ward, 2017*); (ii) the Paleotropical *Tetraponera*, with ~93 species; and (iii) the South American *Myrcidris*, that has only one species described, *Myrcidris epicharis* (*Ward & Downie, 2004*; *Bolton, 2012*; *Ward, 2017*).

According to *Janzen (1966)* and *Ward (1991)*, there are two known Pseudomyrmecinae ecological groups: (i) one composed of generalist arboreal species, and another consisting of (ii) ants specialized in plant colonization. While ants from the Group one nest in dead

sticks of various types of plants and are generally passive in relation to external objects, ants from the Group two are obligate inhabitants of hollow cavities in live tissues (domatia) of plants and are often aggressive towards other insects or plants. Also, ants from Group two provide protection from herbivory and competition to its host plant in a relationship commonly associated with coevolved mutualism (*Janzen, 1966*; *Ward, 1991*).

Previous works using morphological and molecular data (*Ward, 1991*; *Ward & Downie, 2004*) suggest that this kind of mutualism has evolved independently at least 12 times in the Pseudomyrmecinae subfamily. For instance, *Pseudomyrmex* ants evolved similar behaviors by convergence, despite coevolving with different plant hosts (*Ward & Downie, 2004*; *Chomicki, Ward & Renner, 2015*).

Cases of convergent evolution are frequently characterized using phylogenetics approaches (*Ward & Branstetter, 2017*). Evolutionary analyses of mitochondrial sequences often allow a better understanding about the history of taxonomic clades in the level of family (*Miya et al., 2003*; *Kayal et al., 2015*), including inside the subphylum Hexapoda (*Mao, Gibson & Dowton, 2015*; *Bourguignon et al., 2016*). Thus, analyses of mitochondrial genes should be taken on account to study Pseudomyrmecinae, a subfamily that presents several cases of coevolution.

Several molecular studies have been described on *Pseudomyrmex*, generally addressing co-evolutionary questions, such as the impact of mutualistic associations in the rate of genome evolution (*Rubin & Moreau, 2016*) or characterization of ant-plant associations through the study of phylogenetic relationships and biogeography (*Chomicki, Ward & Renner, 2015*; *Ward & Branstetter, 2017*). However, complete mitogenomes analyses have never been performed for Pseudomyrmecinae due to absence of these data. In the current "no budget mitogenomics" approach (defined here as the usage of public raw data to assemble large mitochondrial sequences unavailable at public databases), we used publicly available genomic data generated elsewhere (Table S1) to assemble and analyze the complete mitochondrial sequence for 12 *Pseudomyrmex* and two *Tetraponera* species from Pseudomyrmecinae subfamily. Thus, we present the first dozen of mitogenomes for this subfamily and performed evolutionary analyses on them and all other available Formicidae mitogenomes, trying to better understand the sister clade relationships inside this highly diverse clade. Given the "no budget" nature of this work, the choice of Pseudomyrmecinae species analyzed took advantage of the availability of public data for this clade. The current study presents new complete mitochondrial sequences for ant species that cover five out of 10 *Pseudomyrmex* species groups and almost duplicates the number of mitochondrial genomes available for ants, increasing this number from 15 to 29.

## METHODS

### Data acquisition

Fourteen Illumina paired-end datasets were downloaded from EMBL Nucleotide Archive (https://www.ebi.ac.uk/ena) in SRA file format (see Table S1). The datasets containing both mitochondrial and nuclear data were converted to FASTQ using fastq-dump (with –readids and –split-files parameters) from the SRAtoolkit.2.8.2.

## Mitochondrial genome assembly and annotation

The complete datasets with different number of sequencing reads were used as input for *de novo* assembly using NOVOPlasty2.6.3 (*Dierckxsens, Mardulyn & Smits, 2016*) with default parameters. Since NOVOPlasty was our primary assembler and *Dierckxsens, Mardulyn & Smits (2016)* recommend the use of untrimmed data for this software, we decided to use all datasets without trimming. The only exception was the dataset for *Tetraponera rufonigra*, that had to be trimmed with Trimmomatic v.0.36 (*Bolger, Lohse & Usadel, 2014*) to produce sequences with the same length. This trimming has been performed by setting the parameter MINLEN to match the longest (and modal) read size, therefore discarding shorter reads. NOVOPlasty assemblies needs a seed sequence to start the assembly. Seeds were selected using COX1 (Cytochrome Oxidase I) sequences from the same species (when available) or using COX1 regions from closely-related species. Preliminary mitogenome assemblies by NOVOPlasty were used as reference to a second round of genome assembly using MIRA v.4.0.2 with default parameters (*Chevreux, Wetter & Suhai, 1999*). NOVOPlasty does not generate an alignment file showing the reads mapped to the assembly, so MIRA has been used to map raw sequencing reads to the consensus mitochondrial sequence for the next steps. When the first assembly did not generate the complete mitogenome, we used the largest NOVOPlasty contig as reference for a first mapping assembly using MIRA. The results of this first mapping step were then used as input to MITObim v.1.9 (*Hahn, Bachmann & Chevreux, 2013*) with default parameters, that performed successive iterations to elongate the mitochondrial contig and assemble a circularized version of the mitochondrial genome.

Tablet software version 1.17.08.17 (*Milne et al., 2013*) was used with default parameters to check read coverage and circularization of complete mitogenomes. Automatic annotation performed using MITOS Web Server (*Bernt et al., 2013*) with default parameters and was followed by manual curation using Artemis (*Carver et al., 2012*). The annotation of tRNAs and rRNAs were used in accordance to MITOS Web Server data, except for removing few bases overlapping features in the same strand, when encountered. For the protein-coding genes (PCGs), in many cases, we needed to expand the annotation provided by MITOS Web Server to the closest start codon in order to match the largest Open Reading Frame (ORF) that did not overlap other features in the same strand. Then, we used the online version of BLASTp (*Altschul et al., 1997*) against a database of ants (clade Formicidae) to consider sequence conservation and have information available to decide about the most likely size of the protein, fine-tuning our annotation. Following this procedure, we reached a rational decision on gene boundaries. AT content for (i) the complete mitochondrial genome sequence; and (ii) the intergenic region that contains the D-loop were calculated using the OligoCalc web application (*Kibbe, 2007*).

## Phylogenomics analyses

Formicidae phylogenetics relationships were reconstructed using (i) the 14 complete mitogenomes produced by us together with (ii) all other 15 complete mitochondrial genomes currently available for the clade; and (iii) two mitogenomes of bees (Apidae family) used as outgroups. Two phylogenetics trees have been built using (i) the whole

mitochondrial sequence; and (ii) the concatenated gene set of all 13 protein-coding genes. For the former, we manually edited the sequences to start at the COX1 gene when necessary and aligned the whole mitogenomes using ClustalW version 2.1 using default parameters (*Thompson, Gibson & Higgins, 2003*). For the latter, we aligned and concatenated the nucleotides for all protein-coding genes (PCGs) using the Phylomito script (https://github.com/igorrcosta/phylomito) with default parameters. Modeltest (*Posada & Crandall, 1998*) was run through MEGA7 (*Kumar, Stecher & Tamura, 2016*) with the two datasets and identified the model GTR+G+I as the nucleotide substitution model that better explained sequence variation. Aligned sequences were used as input to a Maximum Likelihood (ML) analysis in MEGA7. Resampling was conducted by bootstrap using 1000 replicates. Blast Ring Image Generator (BRIG) software v.0.95 was run with default parameters (*Alikhan et al., 2011*) to compare and visualize all mitogenomes of Pseudomyrmecinae produced here.

# RESULTS

## Mitogenome assembly and annotation of Pseudomyrmecinae

The 14 genomic datasets used to assemble complete mitogenome sequences for pseudomyrmecine ants were downloaded from SRA database (Table S1). Two different dataset types were used: (i) Whole Genome Sequencing (WGS), that often contained a higher amount of sequencing data totaling 212.7 Gbp (according to SRA information) for six species; an average of 35.45 Gbp per species (*Rubin & Moreau, 2016*); and (ii) UCE experiments, on which we have downloaded 5.94 Gbp for eight species; an average of 742.5 Mbp per species (*Branstetter et al., 2017*; *Ward & Branstetter, 2017*).

The complete dataset downloaded for each species was used as input for a *de novo* sequence assembly using NOVOPlasty. After this first round of genome assembly, we used a subset containing either two or four million sequencing reads as input for a second round of genome assembly using MIRA software. This procedure was performed to both map the sequencing reads into the preliminary assembly and improve the mitogenome quality. For some mitogenomes MIRA could not produce the complete, circularized mitochondrion genome; and a third round of assembly was needed. In that case, the largest contig generated by MIRA has been used as backbone to finish the assembly using MITObim (Table 1). This pipeline was capable to assemble the whole mitochondria of all Pseudomyrmecinae except for *T. aethiops*, on which we have had to use the entire sequencing read dataset for MIRA and MITObim instead of filtering the subset of reads on round 2. The use of multiple strategies to assemble the complete mitochondrial sequences for these species was expected once NGS data is variable amongst different species and sequencing runs; also, the datasets used here came from both different sources and experimental approaches probably potentializing the data variability. The 14 mitochondrial genomes built here were checked for circularity and confirmed to present, as expected for metazoans, 13 protein-coding genes, 22 tRNAs, two rRNAs and a variable control region or D-loop (*Wolstenholme, 1992*; *Prosdocimi et al., 2012*). The genome annotation for all complete mitogenomes is presented (Table S2). All mitochondrial genomes produced here were submitted to GenBank under

**Table 1  Information about mitochondrial genome assemblies of Pseudomyrmecinae.**

| Pseudomyrmecinae species | Species ubrk group | Mitogenome TPA accession number | NOVOPlasty seed | MITObim third assembly round needed | Mitogenome coverage | Low coverage region | Mitogenome size (bp) | AT content: mitogenome (%) | AT content: D-loop region (%) |
|---|---|---|---|---|---|---|---|---|---|
| P. concolor | P. viidus | BK010475 | KU985552.1 | No | 193.2× | No | 15,906 | 75 | 91 |
| P. dendroicus | P. viidus | BK010473 | KP271186.1 | Yes | 123.9× | No | 17,362 | 81 | 94 |
| P. pallidus | P. pallidus | BK010383 | KU985552.1 | No | 91.9× | No | 17,117 | 74 | 84 |
| P. elongatus | P. oculatus | BK010474 | KP271181.1 | No | 115.4× | No | 17,304 | 78 | 93 |
| P. gracilis | P. gracilis | BK010472 | FJ436821.1 | No | 165.5× | 13,761–13,928 | 15,704 | 77 | 93 |
| P. feralis | P. ferrugineus | BK010379 | FJ436819.1 | No | 128.0× | No | 18,835 | 78 | 92 |
| P. ferrugineus | P. ferrugineus | BK010380 | FJ436819.1 | Yes | 87.0× | No | 18,480 | 77 | 90 |
| P. flavicornis | P. ferrugineus | BK010381 | FJ436819.1 | Yes | 152.7× | No | 18,498 | 77 | 90 |
| P. janzeni | P. ferrugineus | BK010382 | FJ436819.1 | No | 125.8× | 15,848–15,867 | 18,380 | 77 | 89 |
| P. particeps | P. ferrugineus | BK010384 | FJ436819.1 | No | 126.8× | 15,799–15,820 | 18,524 | 80 | 90 |
| P. peperi | P. ferrugineus | BK010385 | FJ436819.1 | Yes | 87.4× | 16,006–16,023 | 18,709 | 78 | 91 |
| P. veneficus | P. ferrugineus | BK010386 | FJ436819.1 | No | 155.4× | 15,889–15,928 | 18,410 | 79 | 91 |
| T. aethiops | NE | BK010476 | KX398231.1 | Yes | 712.9× | 13,934–13,982 | 15,988 | 79 | 93 |
| T. rufonigra | NE | BK010387 | KX398231.1 | No | 292.2× | 13,889–13,982 | 15,907 | 74 | 91 |
the Third Party Annotation (TPA) database (*Cochrane et al., 2006*) that provided accession numbers allowing sequence retrieval (Table 1).

According to the average coverage estimate provided by TABLET software, the sequencing read coverage for mitogenomes ranged between 85x and 292x for mitogenomes on which a subset of reads was used. For *T. aethiops,* the coverage was higher as the entire dataset was used (712x). Assembly coverage was observed to be evenly distributed (Fig. S1), except in cases of AT-rich regions that presented low coverage, generally close to poly-T sequences.

## Mitogenome size variation and putative insertion sites in the Pseudomyrmex genus

*Pseudomyrmex* mitogenomes have shown significant variation in size, ranging from 15,704 to 18,835 bp (Table 1). We observed three distinct mitogenome size ranges for the clade (Table 1). Mitogenome size in the genus varied from: (i) less than 16 kb in *P. gracilis* and *P. concolor*; (ii) between 17 kb and 18 kb in *P. pallidus* and *P. dendroicus*; and (iii) higher than 18 kb in other species, that belong to *P. ferrugineus* group. A comparative genomics analysis using BRIG software identified four variable regions as putative insertion segments (Fig. 1). After genome annotation, we identified these presumed insertions to be located between (i) *COX2* and *trn-K*; (ii) *ATP8* and *ATP6*; (iii) *trn-N* and *trn-F*; and (iv) trn-W and *COX1*.

## Gene order arrangements in ant mitogenomes

Regardless of the limited sample of complete mitochondrial genomes analyzed for ants, in general, five slightly different synteny rearrangements (Fig. 2) could be observed in Formicidae family (*Duan, Peng & Qian, 2016*). All Pseudomyrmecinae and Dolichoderinae mitogenomes analyzed showed a single conserved gene arrangement for all species that is also shared by most of Formicinae species. We also observed that Formicinae and Myrmicinae clades present a modal synteny arrangement suggesting a possible ancestral gene arrangement for each group. One single species of Formicinae (*Camponotus atrox*) presents inversions between *trn-M, I* and *Q* that differ from other mitogenomes from this subfamily, possibly representing a derived variation. Myrmicinae also present two other unique rearrangements restricted to a single species each, suggesting derived syntenies: (i) *P. punctatus* has an inversion between *trn-K* and *D*; and (ii) *W. auropunctata* presents both an inversion between *trn-V* and D-loop and a feature (*trnY*) on the opposite strand when compared to the others.

## Phylogenetic analyses of Formicidae using mitogenome data

In order to assess the phylogeny of the group, two Maximum Likelihood trees were produced using slightly different input data: (i) the aligned and concatenated sequences for all 13 mitochondrial PCG's (Fig. 3); and (ii) the complete mitochondrial genomes (Fig. 4). We analyzed all ant species presenting complete mitogenomes available on Genbank (*Gotzek, Clarke & Shoemaker, 2010*; *Hasegawa et al., 2011*; *Berman, Austin & Miller, 2014*; *Babbucci et al., 2014*; *Kim, Hong & Kim, 2016*; *Duan, Peng & Qian, 2016*; *Liu et al., 2016*); *Yang et al., 2015*) and two Apidae bees as outgroups (*Crozier & Crozier,*

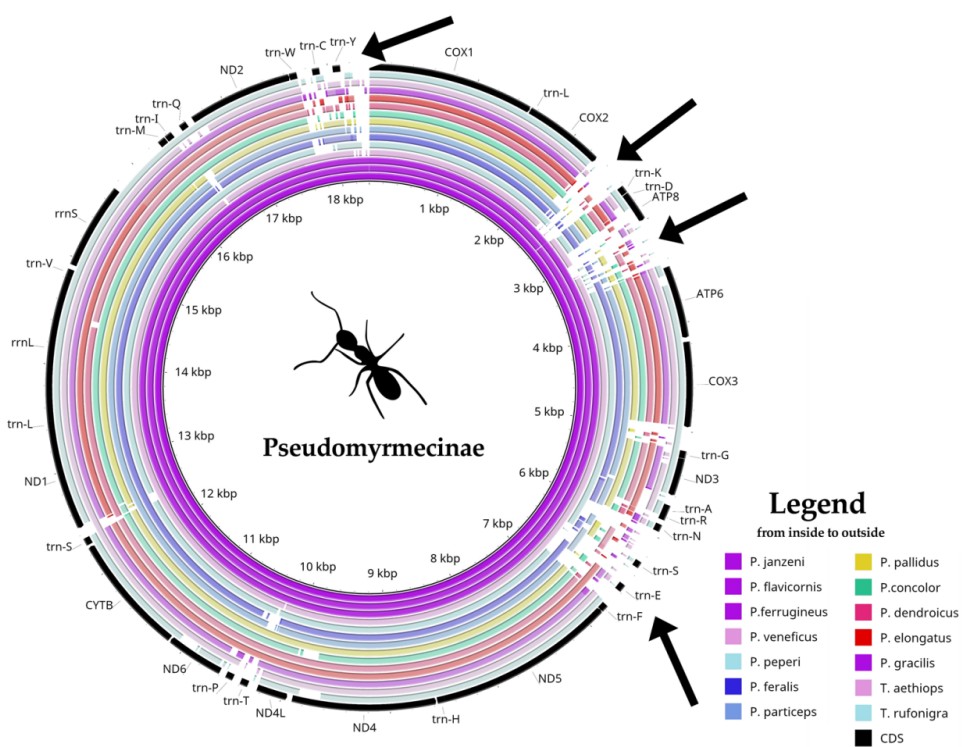

**Figure 1  Comparative genomics analysis of all 14 Pseudomyrmecinae ants.** BLAST comparison of all Pseudomyrmecinae mitochondrial genomes against a reference (*Pseudomyrmex janzeni*) generated by Blast Ring Image Generator (BRIG). Gaps in rings correspond to regions with less than 50% identity to the reference sequence. Most mitochondrial features are conserved within the clade, even though ATP8 and some tRNAs (trn-S, trn-E and trn-T) were observed to be less conserved. Four regions (identified by arrows) present nucleotide size variations between (i) *COII* and *trn-K*; (ii) *ATP8* and *ATP6*; (iii) *trn-N* and *trn-F* and; (iv) *trn-W* and *COI.*

*1993*; *Cha et al., 2007*) (see accession numbers and references for all sequences on Table S3). The trees reconstructed from mitochondrial data corroborated most of the phylogenetic relationships known for ants, with several clades observed as monophyletic with high confidence (bootstrap = 100). Both trees showed similar results, though differences can be observed in several nodes regarding tree topology and/or statistical support. The major difference observed is that the gene-concatenation tree displayed all subfamilies as monophyletic, while Myrmicinae was recovered as paraphyletic in the tree based on complete mitogenomes.

## DISCUSSION

In this study, we used public data to assembly, annotate, compare and provide evolutionary analyses of 14 complete mitochondrial genome sequences from the ant subfamily Pseudomyrmecinae plus 15 other ant mitogenomes downloaded from GenBank.



**Figure 2** **Five different syntenies observed in complete Formicidae mitogenomes available on Genbank.** The two modal gene arrangements are represented inside the horizontal rectangle and were observed in 26 out of 29 species analyzed: all Pseudomyrmecinae (14 species); all Dolichoderinae (two species: *L. pallens* and *L. humile*); three out of four Formicinae (F. fusca, F. selysi and P. dives) and seven out of nine Myrmicinae (*A. texana*; *C. obscurior*; *M. scabrinodis*; *S. richteri*; *S. geminata*; *S. invicta*; *V. emeryi*). We suggest that these may represent the ancestral arrangements for their clades. The syntenies outside the rectangle correspond to unique gene orders encountered in single species. Vertical rectangles and lines indicate regions on which synteny changes occurred, and both the asterisk (*) and the vertical line in the *trn-Y* of *W. auropunctata* indicates that it is the only feature in Formicinae mitochondria that changed its coding strand.

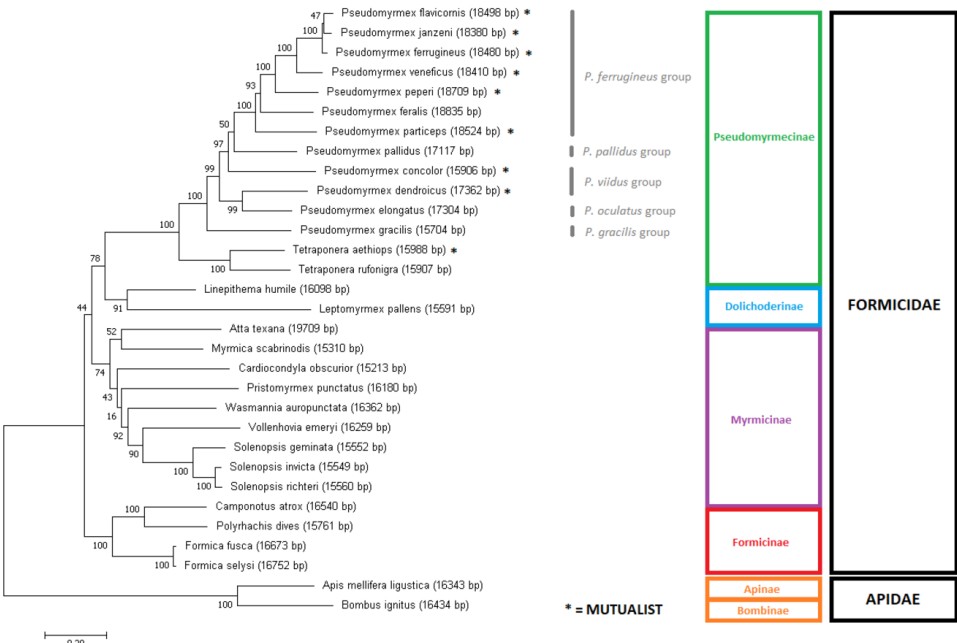

**Figure 3** **Gene-concatenation phylogenomic tree for all Formicidae complete mitogenomes available on Genbank.** The tree was built using the aligned and concatenated nucleotidic sequences for all 13 protein-coding mitochondrial genes. Modeltest identified 'GTR + G + I' as the most adequate substitution model and phylogeny was reconstructed by Maximum Likelihood using MEGA7 software, with 1000 bootstrap replicates. Bees from the Apidae family were used as outgroup. *Pseudomyrmex* species groups are described and mutualistic pseudomyrmecines are evidenced by the presence of an asterisk "*".

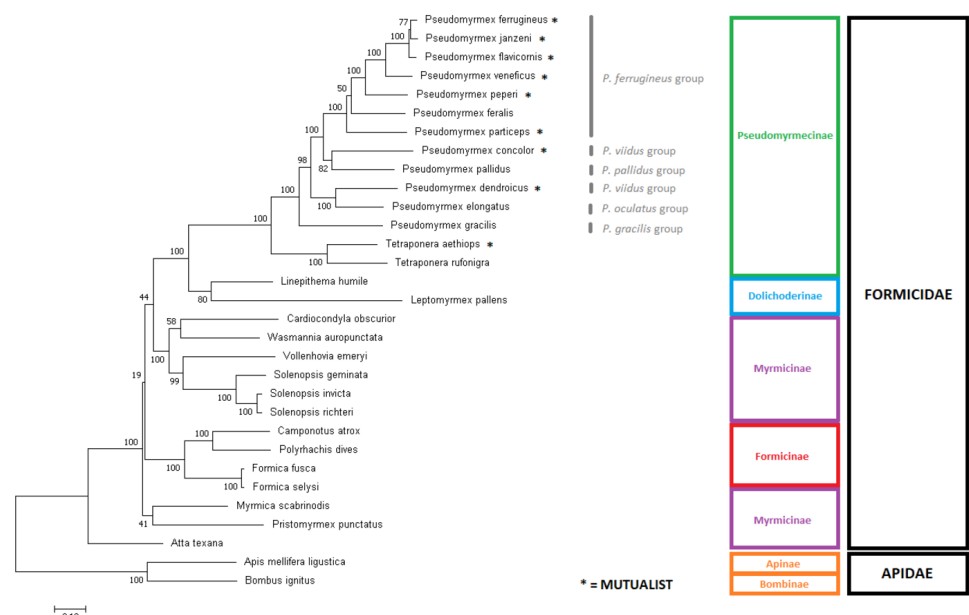

**Figure 4** **Phylogenetic tree using the complete mitochondrial sequence of all complete ant mitogenomes available on Genbank.** 'GTR + G+I' was chosen as substitution model as suggested by Modeltest. The tree was built with MEGA7 using Maximum Likelihood with 1,000 bootstrap replicates. Mitogenomes from bees were used as outgroups. *Pseudomyrmex* species groups and mutualistic pseudomyrmecines are evidenced.

## Uniform genome coverage and expected AT-bias

Even though pieces of the mitochondrion genome may be copied to the nucleus forming NuMTs (Nuclear Mitochondrion Sequences), the genome coverage obtained for the assemblies often presented uniform distributions (Fig. S1), even for *Pseudomyrmex gracilis* on which NuMTs have been previously identified (*Rubin & Moreau, 2016*). The correct assembly of mitochondrial genomes was possible because the number of mitochondrial reads is probably much higher than the number of reads coming from NuMTs.

The low coverage in segments with a pronounced AT-bias should be expected because AT-rich regions are known to have reduced amplification in Illumina library preparation protocols (*Dohm et al., 2008*; *Aird et al., 2011*; *Oyola et al., 2012*). It has been shown that ant mitogenomes have a remarkable AT bias in the D-loop that exceeds 90% (*Berman, Austin & Miller, 2014*; *Liu et al., 2016*). Our data also corroborates this, as 12 out of 14 Pseudomyrmecinae species presents an AT content value that is equal or superior to 90% in the intergenic region between rrnS and trn-M that contains the D-loop (Table 1). Also, this region has already been proved to be particularly difficult to sequence in hymenopterans (*Castro & Dowton, 2005*; *Dowton et al., 2009*; *Rodovalho, 2014*). The difficulties in obtaining the complete mitogenomes for ants could be the reason why there are so few mitochondrial genomes available for this group despite the availability of public data. At the same time, new assemblers like NOVOPlasty outperform classic programs (*Dierckxsens, Mardulyn & Smits, 2016*; *Plese et al., 2018*), facilitating the production of complete mitogenomes. These

advances provide favorable prospects for the closure of the mitogenome phylogenetic gaps in Formicidae, especially if public data is employed for that end.

## Comparative mitogenomics: mitogenome size and synteny analyses

Aside from the identification of four putative insertion sites that could explain the differences observed in mitogenome size (pointed by arrows in Fig. 1), we also observed that all seven mitogenomes included in *P. ferrugineus* group have approximately the same genome size in bp, suggesting that this group is monophyletic. On the other hand, there is a significant difference in mitogenome size between *P. concolor* (15906 bp) and *P. dendroicus* (17362 bp), both belonging to *P. viidus* species group. This corroborates previous works indicating that this species group is paraphyletic (*Ward, 1989*; *Ward & Downie, 2004*).

There is a positive correlation between the multiple syntenies encountered within Myrmicinae and Formicinae clades and the remarkable biodiversity observed for these subfamilies: Myrmicinae is the largest ant subfamily in species richness, with over 6,600 species described, almost half of all biodiversity documented for ants; and Formicidae is the second most biodiverse, featuring over 3,100 species. Other Formicidae subfamilies in this study are not nearly as diverse: Dolichoderinae has ∼713 species while Pseudomyrmecinae presents ∼231 species documented (*Bolton, 2012*). Ancestral gene arrangement for Formicinae is identical to the one observed in Pseudomyrmecinae and Dolichoderinae, signaling that Formicinae is closely related to this group than to Myrmicinae.

A higher number of mitogenomes and broader taxon coverage will improve the assessment of correlation between mitochondrial gene order and subfamily biodiversity, allowing a better understanding of synteny evolution in ant mitochondria.

## Phylogenomic relationships of Formicidae inferred using mitogenome data

The phylogenomic trees generated provided slightly different topologies due to the additional data present in intergenic regions, such as tRNAs, rRNAS and D-loop (*Wolstenholme, 1992*; *Prosdocimi et al., 2012*). It is also known that mitochondrial DNA presents a relatively high substitution rate in non-coding regions (*Vanecek, Vorel & Sip, 2004*; *Desalle, 2017*).

Overall, in the phylogenomic trees generated for the whole Formicidae family, the phylogeny of the subfamily Pseudomyrmecinae was strongly retained as monophyletic, and the phylogenetic positions of most clades were well resolved. The monophyly for the Pseudomyrmecinae subfamily and also for *Pseudomyrmex* and *Tetraponera* genera were recovered with 100% bootstrap support (BS) in both trees. The genus *Pseudomyrmex* presented few unsupported nodes, but *Tetraponera* was completely resolved on both trees (BS = 100). In both trees, both the monophyletic status of the *P. flavicornis* group and the paraphyletic status of the *P. viidus* group confirms (i) previous observations based exclusively on morphology (*Ward, 1989*), (ii) phylogenies using both morphological characters and few nuclear markers (*Ward & Downie, 2004*), and (iii) our own observations regarding mitogenome size. Although the morphological division in species groups has not been formalized or regulated under nomenclatures (*Ward, 2017*), the work using a

hybrid morphological/molecular approach of *Ward & Downie (2004)* shows that only two out of nine groups defined at the time were paraphyletic: *P. pallens* and *P. viidus* groups. The corroboration of morphological studies by mitochondrial data analysis confirms the relevance of using morphological characters in determining relationships between clades, but also reinforces that molecular evidence can clarify and complement such studies, refining and improving the overall support of the phylogenies reconstructed. Under this work, we generated complete mitochondrial sequences for ants representing five out of the 10 groups described for *Pseudomyrmex* species, covering at least half of *Pseudomyrmex* genetic diversity and adding a new source of molecular evidence for further studies on the clade.

Both trees suggest strongly that ant-plant mutualisms are paraphyletic in Pseudomyrmecinae (please check asterisks in Figs. 3 and 4), also adding evidence to previous assumptions of generalist behavior as a basal trait in the *Pseudomyrmex* genus (*Ward & Branstetter, 2017*). This suggests that ant-plant coevolution developed later (and independently) several times in the clade. Mutualistic species are more common in the *P. ferrugineus* species group, strengthening the hypothesis of mutualism being a derived trait in Pseudomyrmecinae. In the *Pseudomyrmex* genus, the *P. ferrugineus* group features may present two independent lineages of mutualistic ants (considering that *P. feralis* is often considered to display generalist behavior; BS = 50), while other two independent mutualistic lineages can be observed by the phylogenetic placement of the species *P. concolor* and *P. dendroicus*. Considering the *Tetraponera* genus, *T. aethiops* and *T. rufonigra* are closely related species and only *T. aethiops* presents exclusive ant-plant mutualistic behavior. This shows that evolution of mutualistic traits in Pseudomyrmecinae may have occurred in close related species. So, considering the limited number of species sampled here, we were able to identify five out of the 12 times that mutualistic associations have been reported to appear in the clade (*Ward, 1991*; *Ward & Downie, 2004*). With a better taxonomic coverage, this number can be increased and new analyses performed, further improving our understanding about these coevolutionary events.

Well resolved relationships for several Pseudomyrmecinae species (such as *P. peperi*, *P. veneficus*, *P. particeps*, *P. gracilis*, *T. aethiops* and *T. rufonigra*) corroborate both the results of *Ward & Downie (2004)* and the ML tree generated using UCE data from *Ward & Branstetter (2017)*. The sister group relationship between *P. dendroicus* and *P. elongatus* is also well supported (BS = 100 in complete mitochondria tree; and BS = 99 in gene-concatenation tree), in line with a recent work using concatenated WGS scaffolds as input for ML tree reconstruction (*Rubin & Moreau, 2016*).

However, subtle differences were observed between our results and the inferred UCE multiloci phylogenetic relationships (*Ward & Branstetter, 2017*). Using UCE data, *P. janzeni* was observed as sister group to *P. ferrugineus.* Here, the complete mitogenome tree recaptured this same relationship with a bootstrap replicate value of 77. On the other hand, in the concatenated gene set, the sister group relationship observed between *P. janzeni* and *P. flavicornis* showed a lower support (BS = 47). Overall, this relationship seemed to be better resolved by the analysis of the complete mitochondrial sequence, also corroborating the UCE analyses.
Both trees showed Dolichoderinae subfamily as monophyletic, even though this result was not recovered in all replicates. Dolichoderinae is a highly diverse subfamily and contains over 700 species, but it has been represented here by merely two species. Thus, we believe that a higher coverage of species will improve the robustness of the phylogenetic analyses.

Previous work with morphological characters and/or nuclear genes presents evidence of sister group relationship between Pseudomyrmecinae and Myrmeciinae (*Ward & Downie, 2004*; *Brady et al., 2006*). We also expected Myrmeciinae to be sister group to Pseudomyrmecinae according to mitochondrial data, but once complete mitochondrial genomes are not available for the subfamily Myrmeciinae we could not test this hypothesis. The complete absence of mitogenomes for this and other subfamilies might lie in the fact that their diversity is not expressive in comparison to the species richness of the most diverse, sampled and studied ant subfamilies. While Myrmeciinae has only 94 described species, Myrmicinae has over 6,600 species (*Bolton, 2012*). In the absence of Myrmeciinae, Dolichoderinae is expected to be the closest relative to Pseudomyrmecinae in our trees, which occurs in, both trees corroborates large-scale molecular phylogenies using few nuclear genes (*Brady et al., 2006*) and UCE data (*Branstetter et al., 2017*). Shared synteny between all Pseudomyrmecinae and Dolichodrinae sampled also supports the sister group relationship observed. Our results support the evidence that Myrmicinae as sister taxa to a clade containing both Pseudomyrmecinae and Dolichoderinae, while Formicinae has been observed as a more basal group in the Formicidae family. This position for Formicinae is highly supported in the gene-concatenation tree but not in the tree using complete mitogenomes. This position is not supported by other works using nuclear data that supports the evidence of a sister group relationship between Myrmicinae and Formicinae (*Brady et al., 2006*; *Branstetter et al., 2017*).

The monophyly of the subfamily Formicinae and all its nodes show maximum support on both trees (BS = 100). These trees also confirm the monophyly for the genus *Formica* and show genera *Camponotus* and *Polyrhachis* as closely related to each other, as observed in the work of Blaimer and collaborators (*2017*) that used UCE loci for tree inference. The only issue in this subfamily concerns the unsupported phylogenetic placement of Formicinae in relation to other subfamilies. Mitogenome data successfully delivered sound phylogenetic relationships even for *Camponotus atrox* that showed a unique synteny but have had its position well resolved in both trees, including in the complete mitochondrial tree, that may be prone to suffer from synteny changes. This issue confirms the robustness of mitochondrial sequences to infer ant phylogenies.

Overall, the most controversial results obtained here are related to the position of the subfamily Myrmicinae. For that clade, the gene concatenation tree was capable to indicate monophyly (BS = 74) but whole mitogenome data produced paraphyly. In the latter case, the myrmicine ants *Atta texana*, *Myrmica scabrinois* and *Pristomyrmex punctatus* derived earlier than the other ants. On the other hand, some relationships were recovered with 100% bootstrap support, such as the monophyly of the genus *Solenopsis*. Our results corroborate those obtained by the use of concatenated amino acid sequences of all mitochondrial PCGs for tree inference (*Duan, Peng & Qian, 2016*). However, our assessment of the position of *V. emeryi* was better supported (BS = 90 on gene-concatenation tree and BS = 99 on

complete mitochondria tree) than that of this previous work (BS = 75). Considering that Duan and collaborators (2016) used a similar approach to ours (gene concatenation under a Maximum Likelihood method), we may conclude that these better results indicate that nucleotide data presents more reliable information for these clades than amino acid data. This is consistent with previous comparative work in which tree inference at nucleotide level has outperformed amino acid and codon analyses (*Holder, Zwickl & Dessimoz, 2008*). Moreover, bootstrap values obtained from nucleotide data have been reported to be often higher than their amino acid correspondents (*Regier et al., 2010*). This observation is at least partly explained by the differences in the amount of phylogenetic signal considered by these two methods. Additional signal present in nucleotide data is missed when they are translated into amino acids. This is particularly important when hexacodonic amino acids are considered; serine, for instance, is encoded by TCN and AGY (*Regier et al., 2010*; *Zwick, Regier & Zwickl, 2012*).

In both works, mitogenome analyses were not fully capable of resolving important nodes of the myrmicine branch and several factors may be involved in these unsatisfactory results. It is necessary to highlight that Myrmicinae is the most biodiverse ant subfamily (*Bolton, 2012*) and it is known to feature several dubious monophyletic groups (*Brady et al., 2006*; *Ward, 2011*; *Ward et al., 2015*). This diversity is evidenced by the fact that, despite only nine mitogenomes are available for the group, three different mitochondrial gene arrangements can be observed, suggesting a high rate of mitochondrial evolution in this subfamily.

Also, there have been divergences in the Myrmicinae branch of previous molecular phylogenetic studies attempting to study the Formicidae family (*Brady et al., 2006*; *Moreau et al., 2006*). On the other hand, *Ward et al. (2015)* focuses on the subfamily by reconstructing a large-scale phylogeny using 11 nuclear markers from 251 species sampled across all 25 myrmicine tribes, most of them nonmonophyletic. By using such huge amounts of data covering a great part of Myrmicinae species diversity, they managed to propose a new classification of Myrmicinae consisting of exclusively monophyletic tribes, which also reduced the number of genera that are not monophyletic.

Thus, the hyperdiverse nature of this clade, associated to poor taxon sampling and a possible high rate of mitochondrial genome evolution may have contributed to produce inconclusive results in mitochondrial analyses. Also, even though some relationships were not elucidated by mitochondrial phylogenomics alone, the information provided by the mitogenome has been proven several times to be useful in the study of evolutionary relationships for several taxa, either confirming (*Prosdocimi et al., 2012*; *Finstermeier et al., 2013*) or refuting previous phylogenetic hypotheses (*Kayal et al., 2015*; *Uliano-Silva et al., 2016*). Therefore, we still recommend the use of mitochondrial data, preferably alongside other markers (i.e., nuclear genes), to increase phylogenetic signal and recapture phylogenies. However, due to the substitution rate of mtDNA, trees generated from mitochondrial data have higher probability of resolving short internodes correctly (*Desalle, 2017*). Thus, we also believe that mitochondrial data alone will yield better results for this and other branches if we address the shortage of mitogenomes available for this clade by improving

mitochondrial taxon coverage and reducing tree internodes. In that sense, results present here are extremely relevant to show that information already available in public databases should be used to obtain such sequences at no additional sequencing costs.

## No budget mitogenomics: integrative analyses between datasets and potential for large-scale studies

The results presented here confirm that both UCE and WGS data publicly available can be used to assemble complete mitochondrial genomes with high coverage (Table 1), which can be explained by the high copy number of mitochondrial genome reads compared to nuclear genomes sequencing reads that may reach something between 0.25% to 0.5% of the total number of bases generated (*Prosdocimi et al., 2012*), sometimes reaching percentages as high as 2% of reads mapping to mtDNA (*Ekblom, Smeds & Ellegren, 2014*). We also confirm the potential of UCE data as a low-cost alternative to sequence complete mitogenomes with high coverage as described by *Raposo do Amaral et al. (2015)*. Mitogenome data is used in various types of analyses and mitochondrial sequences are encountered in several types of datasets, normally providing enough information to assemble the entire mitochondrial sequence. This versatility and ubiquity of mitogenome information should be used in favor of biodiversity studies, especially considering the increasingly available public datasets for a great number of species.

The potential of these sequences in unveiling phylogenies must not be overlooked, especially if we consider that there are different dataset types available for different species (WGS, RNA-Seq, UCE enrichment, among others). These different resources makes it difficult to achieve an integrated phylogenetic/phylogenomic analyses using the public data, that often depends on sequence orthology to be performed (*Kuzniar et al., 2008*). Thus, the use of different types of data to assemble the complete or nearly complete mitogenomes for species with publicly available data presents a solution to this problem, with the mitochondrial genome acting as a "normalizing sequence" that allows the comparison of different datasets. For instance, in this work some species had only UCE data publicly available, while others presented standard WGS datasets. Yet, by assembling, annotating and analyzing the complete mitogenome for these species, we were able to broaden our scope and study all of them together. Thus, we suggest that the use of mitogenomes obtained from public data has the potential to become an important source of phylogenetic information. Besides, the study of mitochondrial sequences may be one of the fastest routes towards a high-quality comprehensive species-level tree for hyperdiverse taxa such as insects. Steps have been taken that way, as it can be seen on recent work by *Linard et al. (2018)*, where data mining from Genbank and assembly of metagenomic datasets provided mitochondrial contigs (>3 kbp) for almost 16,000 coleopteran species. This huge amount of data was used to generate the largest phylogenetic tree for the clade.

Studies that attempt to assemble complete mitogenomes using public data are yet scarce whereas the size and breadth of public databases is ever growing, along with its potential to answer phylogenetic questions. No budget mitogenomics represents an unprecedented opportunity to reconstruct and analyze large-scale phylogenies for various groups at

different taxa levels, which in turn may help other evolutionary and conservation biology studies and promote an overall increase on our knowledge about non-model species and their diversity.

## CONCLUSION

Here we assembled and annotated the first 14 mitogenomes for the ant subfamily Pseudomyrmecinae using a pipeline that relies solely on public data from different sources and types, making profit of bioinformatics software publicly available. These sequences were used to study synteny, comparative genomics and phylogenomic analyses providing valuable information regarding Pseudomyrmecinae phylogeny and evolution, such as: (i) identification of four putative mitochondrial indel sites in *Pseudomyrmex*; (ii) corroboration that mutualistic associations independently arose in the clade many times; (iii) indication that *P. ferrugineus* group is monophyletic and *P. viidus* species groups is paraphyletic; and (iv) corroboration of monophyletism for the *Pseudomyrmex* and *Tetraponera* genera. Mitochondrial data on other ant clades, though limited, were useful in both synteny and phylogenomic analyses to broaden our scope and allow the study other ant groups. This allowed us to unveil sister group relationships throughout the family, such as the one between Pseudomyrmecinae and Dolichoderinae; as well as the monophyletic status of all subfamilies analyzed. However, a more precise definition about the relationship among groups should make use of large genome datasets and gene concatamers built with hundreds to thousands of genes; currently unavailable. Besides, low bootstrap values observed at some nodes, indicate that mitochondrial data available do not present enough variability to elucidate some relationships. The mitochondrial sequences assembled cover a considerable portion of Pseudomyrmecinae biodiversity and will be useful for further evolutionary and conservational studies. This work practically doubles the number of complete ant mitogenomes available at no additional sequencing costs. Since mitogenome taxon coverage is still lacking for Formicidae, its improvement is desirable for better resolution and robustness of large scale phylogenies in the group. This pipeline can also be used to study the aforementioned dubious monophyletic clades in Myrmicinae (*Brady et al., 2006*; *Ward, 2011*; *Ward et al., 2015*) or paraphyletic groups, such as the *Camponotus* genus from the Formicinae subfamily (*Blaimer et al., 2015*). Based on these results, we emphasize that the ever-increasing breadth of public databases, associated to the possibility of obtaining mitochondrial sequences from different types of sequencing data makes no budget mitogenomics the ideal approach for the study of species diversity and, possibly, the fastest route toward species-level phylogenetic trees.

### Funding

This work was supported by CAPES and Fundação de Amparo a Pesquisa do Estado do Rio de Janeiro (FAPERJ) (grant numbers 202.810/2015 and 202.780/2018). The funders had

no role in study design, data collection and analysis, decision to publish, or preparation of the manuscript.

## Grant Disclosures

The following grant information was disclosed by the authors:

CAPES.

Fundação de Amparo a Pesquisa do Estado do Rio de Janeiro (FAPERJ): 202.810/2015, 202.780/2018.

## Competing Interests

The authors declare there are no competing interests.

## Author Contributions

- Gabriel A. Vieira and Francisco Prosdocimi conceived and designed the experiments, performed the experiments, analyzed the data, contributed reagents/materials/analysis tools, prepared figures and/or tables, authored or reviewed drafts of the paper, approved the final draft.

## DNA Deposition

The following information was supplied regarding the deposition The following information was supplied regarding the deposition of DNA sequences:

GenBank accession numbers BK010472–BK010476 and BK010379–BK010387.

The complete sequence for all assembled mitogenomes is provided in File S1.

## Data Availability

The direct links for download of all 14 Pseudomyrmecinae genomic public datasets (in SRA file format) used for mitogenome assembly are provided in Table S1.

## Supplemental Information

Supplemental information for this article can be found online at http://dx.doi.org/10.7717/peerj.6271#supplemental-information.

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
