# Peer review of "Accessible molecular phylogenomics at no cost: obtaining 14 new mitogenomes for the ant subfamily Pseudomyrmecinae from public data"

_PeerJ, doi:10.7717/peerj.6271_

## Round 0.1 · original submission · Minor Revisions

Both reviewers were positive about this manuscript and both of them have taken the time to provide what I think are very detailed and helpful critiques. One reviewer has provided an annotated version of the manuscript. Please think carefully through all the feedback that has been provided and do your best to incorporate edits that address these points.

·

Basic reporting

ENGLISH: English is clear, at least for another native portuguese speaker. However, I could identify some typos throughout the MS. For example, 1 - on the fourth line of the Abstract authors write "... amount of genome data ..." rather than "amount of genomic data"; 2 - at the end of the Abstract "Formcidae" [missing i] rather than "Formicidae"; 3 - 1 to 9 are frequently written as numerals; 4 - also pay attention to prepositions and conjunctions, as in line 188 where authors used "... the coverage was higher once the entire ..." and it should be "higher as the entire".

ARTICLE STRUCTURE: In Figure 1 Cytochrome oxide I and II are referred as COI and COII in the caption but as cox1 and cox2 in the figure. Should be harmonised. The caption of Figure 2 states “Synteny of all complete mitogenomes available on Genbank”, but only five mitogenomes are shown. From the main text, it is easy to infer those five mitogenomes are representative from those available in Genbank for the family, but caption should be more precisely written. Also the caption of Figure 2 says “arrow(s)”, but there are lines and not arrows in the figure.

Experimental design

METHODS: Line 128 – Authors say “… except by the dataset for Tetraponera rufonigra, that had to be trimmed […] to produce sequences with the same length”. This is not totally clear. The dataset for the other species were not trimmed? Maybe a little more background on the used datasets would be elucidative. Moreover, should not the datasets for all species be trimmed for their quality (to guarantee, for example, PHRED > 30)?

Validity of the findings

Line 261 - "P. ferrugineus group have approximately the same genome size in bp, suggesting that this group is monophyletic.”. Genome size per se is not suggestive of phylogenetic proximity or relationship.
Line 290 – “(iii) our own observations regarding mitogenome size.”. Same comment as above.
Line 365 – “In the later case [tree using the complete mitochondrial sequence], a clade consisting of myrmicine ants Myrmica scabrinois, Pristomyrmex punctatus and Atta texana appear as basal group”. However, these three species does not form an clade in the correspondent figure, as they do not share an exclusive common ancestor.

Additional comments

The MS #32218 reanalyses raw reads from high throughput DNA sequencing deposited in public databases to make more sense from this freely available information, to increase the amount of structured genetic information about an ant family and to contribute to the understanding of their evolution and ecological habits. In the opinion of this reviewer it was performed in accordance with the standard methods used in the field, it has merit and, therefore, should be accepted for publication. Before final acceptance, however, the points commented in the first three sections should be addressed. In addition, I also suggest: 1 - line 80, include "ecological" in "There are two known groups ..."; 2 - Line 252, "... AT bias in the control region that exceeds 90%", authors could provide the percentage of AT in the control regions of the newly assembled mitogenomes; and 3 - in the discussion, authors should consider to give more emphasis to the "comparative mitogenomics" rather than to the "Phylogenomic relationship of Formicidae inferred using mitogenome data". Note that the "comparative mitogenomics" covers half of a page while the "Phylogenomic relationship of Formicidae inferred using mitogenome data" spams more than five pages. The main contribution of this paper is the "Assembling 14 new mitogenomes", as reflected in the title, but not in the discussion.

Reviewer 2 ·

Basic reporting

All information are in the pdf that I attached.

Experimental design

All information are in the pdf that I attached.

Validity of the findings

All information are in the pdf that I attached.

Additional comments

All information are in the pdf that I attached.

Annotated reviews are not available for download in order to protect the identity of reviewers who chose to remain anonymous.

---

## Round 0.2 · accepted · Accept

Thank you for carefully addressing all the reviewers' comments.

#